# Stem-Cell Therapy for Bronchopulmonary Dysplasia (BPD) in Newborns

**DOI:** 10.3390/cells11081275

**Published:** 2022-04-09

**Authors:** Said A. Omar, Amal Abdul-Hafez, Sherif Ibrahim, Natasha Pillai, Mohammed Abdulmageed, Ranga Prasanth Thiruvenkataramani, Tarek Mohamed, Burra V. Madhukar, Bruce D. Uhal

**Affiliations:** 1Division of Neonatology, Department of Pediatrics and Human Development, College of Human Medicine, Michigan State University, East Lansing, MI 48824, USA; abdulhaf@msu.edu (A.A.-H.); ibrahi22@msu.edu (S.I.); natasha.pillai@hotmail.com (N.P.); abdulmag@msu.edu (M.A.); thiruve5@msu.edu (R.P.T.); mohame54@msu.edu (T.M.); madhukar@msu.edu (B.V.M.); 2Regional Neonatal Intensive Care Unit, Sparrow Health System, Lansing, MI 48912, USA; 3Department of Physiology, Michigan State University, East Lansing, MI 48824, USA; uhal@msu.edu

**Keywords:** bronchopulmonary dysplasia, stem cells, hyperoxia, lung injury, extracellular vesicles, premature infants

## Abstract

Premature newborns are at a higher risk for the development of respiratory distress syndrome (RDS), acute lung injury (ALI) associated with lung inflammation, disruption of alveolar structure, impaired alveolar growth, lung fibrosis, impaired lung angiogenesis, and development of bronchopulmonary dysplasia (BPD) with severe long-term developmental adverse effects. The current therapy for BPD is limited to supportive care including high-oxygen therapy and pharmacotherapy. Recognizing more feasible treatment options to improve lung health and reduce complications associated with BPD is essential for improving the overall quality of life of premature infants. There is a reduction in the resident stem cells in lungs of premature infants with BPD, which strongly suggests a critical role of stem cells in BPD pathogenesis; this warrants the exploration of the potential therapeutic use of stem-cell therapy. Stem-cell-based therapies have shown promise for the treatment of many pathological conditions including acute lung injury and BPD. Mesenchymal stem cells (MSCs) and MSC-derived extracellular vesicles (EVs) including exosomes are promising and effective therapeutic modalities for the treatment of BPD. Treatment with MSCs and EVs may help to reduce lung inflammation, improve pulmonary architecture, attenuate pulmonary fibrosis, and increase the survival rate.

## 1. Introduction

Preterm birth is a progressively increasing issue among newborn babies, with about 14.9 million babies born before 37 weeks of gestation worldwide; this accounts for approximately 11% of all neonates. About 50% of all newborn deaths are because of early gestational age-related complications [1]. Because of medical and technological advancements, survival rates for preterm babies are also increasing at a fast pace. However, these newborns are more susceptible to a wide range of short-term and long-term complications. Premature newborns who initially have minimal or no lung disease develop increasing oxygen and ventilatory needs over the first several weeks of life. Preterm babies may suffer from respiratory failure because of different causes such as respiratory distress syndrome (RDS) and acute lung injury (ALI).

Premature infants with these respiratory disorders usually require mechanical ventilation that can lead to tissue damage [2]. The prolonged use of assisted ventilation can lead to high oxygen level (hyperoxia) in the lungs. Hyperoxia is thought to cause inflammation that can lead to abnormal lung development. Hyperoxia disrupts vascular and alveolar growth of the developing lung and contributes to the development of bronchopulmonary dysplasia (BPD) [3]. Therefore, premature infants treated with supplemental oxygen and mechanical ventilation have a higher chance of developing BPD. BPD is the most common form of chronic lung disease that affects premature babies and contributes to their morbidity and mortality [4]. The risk of developing BPD increases with decreasing birth weight and gestational age [5]. BPD develops in approximately 25% of infants with a birth weight under 1500 g [6]. Preterm babies that survive BPD have a higher risk of persistent respiratory problems, due to the lack of available treatment for the disease [1]. Neonates with BPD may experience various long-term lung complications such as asthma, respiratory infections, low exercise capacity, and early-onset emphysema [7,8,9,10,11,12].

## 2. Mechanism of Lung Injury in Premature Newborns

The injury resulting in BPD likely begins as altered lung development before delivery in many infants (small for gestational age, chorioamnionitis, tobacco exposure); it can be initiated by resuscitation at birth, and then amplified by postnatal exposures (high oxygen exposure, mechanical ventilation associated with barotrauma, volutrauma, and infection) [13].

Inflammatory stimuli such as sepsis, chorioamnionitis, and hyperoxia have been shown to disrupt growth factor signaling, extracellular matrix formation, and cell proliferation in developing lungs and contribute to BPD pathogenesis. The primary pathology in BPD is the suspension of alveolarization postnatally and a decrease in type I alveolar epithelial cells [14]. Prolonged hyperoxia is associated with lung injury and development of alveolar simplification [15].

Premature newborns are at a higher risk of sepsis and mortality secondary to sepsis than other age groups [16]. Infection induced by LPS exposure in neonatal mice or chorioamnionitis and hyperoxia are associated with delay rodent lung development, especially in the saccular stage, through modulating the expression of different cytokines and chemokines. LPS predominantly affects IL-1β mRNA expression, which has been found to be implicated in the pathogenesis of BPD, while hyperoxia affects chemokine (C–C motif) ligand 2 (CCL2) and intercellular adhesion molecule 1 (ICAM1) expression, which leads to disruption of alveolarization in rodents [17,18]. In addition, chorioamnionitis with a fetal inflammatory response is associated with the increased risk of developing BPD [15]. Interestingly, an IL-1 receptor antagonist (IL-1Ra) decreases the detrimental effects of neonatal hyperoxia on murine alveoli and lung vasculature [19].

In a recent study involving a double-hit BPD animal model, the transcriptome analysis of all pulmonary immune cells showed significant upregulation of genes implicated in chemokine-mediated signaling and immune cell chemotaxis and downregulation of genes implicated in multiple T lymphocyte functions [20]. Moreover, prolonged hyperoxia is associated with defective development of T cells in the thymus and altered distribution of T-cell subpopulations [15]. Oxidative stress with increased production of free oxygen radicals secondary to hyperoxia disrupts normal lung development and contributes to the development of BPD [21].

Although mechanical ventilation is often essential and lifesaving, it can provoke ventilator-induced lung injury in very premature infants mainly by overstretching of the distal epithelium and capillary endothelium [22,23,24,25,26]. The development of injury is dependent on the developmental stage of the lung, and the type, duration, volume, and pressure of mechanical ventilation [23,25]. Mechanical ventilation also results in downregulation of vascular endothelial growth factor 1 (VEGF-1) and its receptor VEGFR-1 (flt-1) and upregulation of the transforming growth factor beta (TGF-β) coreceptor Endoglin. This imbalance in mechanically ventilated lungs likely contributes to altered alveolarization and angiogenesis [27,28,29].

## 3. Limitation of the Current Therapy for BPD

The current therapy for BPD is limited to moderately active drugs such as caffeine, and vitamin A or dexamethasone, which have been shown to be associated with severe long-term neurodevelopmental adverse effects [30]. Recognizing more feasible treatment options to improve lung health and reduce complications associated with BPD is essential for improving the overall quality of life of preterm infants. The limitations of pharmacotherapy for BPD have prompted the search for other types of therapeutic modalities. One alternative is stem-cell-based therapy that has shown promise for the treatment of many pathological conditions. In this review, we introduce the most advanced trends in the use of mesenchymal stem cells as a promising treatment for the prevention and control of BPD. These trends include safer and more effective modes of administration that have been tried experimentally, as well as in animal studies and clinical trials, such as the intratracheal route. They also include the use of extracellular vesicles (EVs) as a safer and easier to store and administer therapeutic option which could be as effective as the MSCs while avoiding some of their complications.

## 4. The Rationale for Stem-Cell Therapy for BPD

Preclinical data strongly support the role of progenitor cells in the preservation of lung structure. Stem-cell-based therapy is a new and promising prevention and treatment method for BPD. Adult human stem cells have been found to be naturally capable of maintaining, generating, and replacing terminally differentiated cells. Stem cells function in response to physiologic cell turnover or tissue damage due to injury as caused by mechanical ventilation, barotrauma, volutrauma, and hyperoxia in premature babies [2,31].

### 4.1. Changes in the Number of Stem Cells in Adult and Neonatal Lung Injury

A study of human adult patients with acute lung injury (ALI) demonstrated that endothelial progenitor cells (EPCs) are mobilized in the circulation, and that the number of circulating EPCs in patients with ALI is approximately twofold higher than in healthy control subjects. An increased number of circulating EPCs was associated with improved survival [32]. These data suggest that EPCs play a significant role in the repair of lung injury. As adult ALI is different from BPD, another study used neonatal mice to explore whether hyperoxia contributes to abnormalities in lung structure by impairing EPC mobilization and homing to the lung in comparison with adult mice [3]. This study found that even moderate hyperoxia decreases vessel density, impairs lung structure, and reduces EPCs in the circulation, bone marrow, and lung of neonatal mice but increases EPCs in adults. This developmental difference between neonates and adults may be owed to the increased susceptibility of the developing lung to hyperoxia and may contribute to impaired lung vascular and alveolar growth in BPD [3].

### 4.2. Role of Nitric Oxide (NO) and EPCs in Promoting Angiogenesis in Hyperoxia-Induced Lung Injury

Other than the sole idea that reduced circulating EPCs leads to a higher risk of developing BPD in neonates, some research studies have data suggesting that impaired nitric oxide (NO) production may contribute to the pathogenesis of BPD [33,34], and that administration of NO can improve lung vasculature and structure in neonates or BPD models [35,36,37]. NO is thought to promote angiogenesis via the synthesis of vascular endothelial growth factor (VEGF) [37]. One study using endothelial progenitor cells demonstrated that endothelial nitric oxide synthase (eNOS) is associated with increased mobilization of EPCs and angiogenesis, suggesting that NO is required for EPC-induced angiogenesis [38]. NO contributes to important homing factors such as differentiation, survival, and adhesion of EPCs. Inhaled NO (iNO) is known to be able to downregulate NF-κB transcription factors [39], reduce expression of proinflammatory cytokines, inhibit leukocyte trafficking in lungs, release inflammatory mediators [40], protect barrier function of the alveolar–capillary membrane [41], and increase the antioxidant capacity of the lungs [42]. Endothelial nitric oxide synthase (eNOS) has been demonstrated to be associated with increased mobilization of EPCs and angiogenesis, suggesting that NO is required for EPC-induced angiogenesis [38]. One study found that ex vivo generated EPCs could home to hyperoxic lungs of neonatal mice, and that iNO improved the engraftment of transplanted EPCs in hyperoxic lungs. When EPC treatment was combined with iNO administration, the lung airspaces and vascularity were significantly improved, suggesting that EPC transplantation and iNO can have a synergistic effect in hyperoxic lungs [43].

### 4.3. Role of Lung-Resident Stem/Progenitor Cells in the Development of BPD

Lung-resident stem/progenitor cells include cells of endothelial, mesenchymal, and epithelial lineages [44]. Lung epithelial stem/progenitor cells, like other stem cells, are capable of giving rise to differentiated cell lineages. In a study on neonatal rats [45], trans-differentiation of type II alveolar epithelial cells (AT2) into type I alveolar epithelial cells (AT1) was, not surprisingly given their normal functions as stem cells in lung injury [46], found to be increased under hyperoxic treatment. However, such repair during injury cannot offset pulmonary epithelial air exchange and barrier dysfunction caused by structural damage to alveolar epithelial cells [45]. Alveolar septation and angiogenesis in the developing lung is regulated by lung-resident mesenchymal stem/stromal cells (L-MSCs). L-MSCs are stem cells found within the lung mesenchyme differentiating to daughter cells including airway smooth muscle cells or stalk mesenchyme fibroblasts [47,48,49]. In a study that included human fetal and neonatal rat lungs, endothelial progenitor cells were found to be involved in vascular repair and include a subset with intrinsic self-renewal potential called endothelial colony-forming cells (ECFCs) [50]. ECFCs were found to be lower in the cord blood of human infants with BPD, while those with high levels of ECFCs were protected from developing BPD [51,52]. Circulating ECFCs from preterm human neonates were shown to be highly susceptible to hyperoxia in vitro, impairing their functionality. This shows the vital contribution of endothelial progenitors to the disruption in lung vascular growth and homeostasis in infants with BPD [53]. Thus, it is hypothesized that functional impairment or depletion of these lung-resident stem/progenitor-cell populations contributes to the disease pathogenesis in BPD [44,48].

## 5. Animal Studies on Mesenchymal Stem-Cell Therapy for BPD

As mentioned earlier, several studies found a reduction in resident stem cells in the lungs of BPD infants and strongly suggested a critical role of stem cells in BPD pathogenesis, warranting their potential use as a treatment approach [44,51,52,54]. Mesenchymal stem/stromal cells (MSCs) have been widely investigated as a potential tool for preventing and treating many lung diseases. MSCs were originally discovered in the bone marrow, and criteria for identifying them focused on the presence of cell-specific markers CD105, CD90, and CD73, the absence of surface markers CD45, CD34, CD14, CD11b, CD79a, CD19, or HLA-DR, adherence to plastic, and the ability for in vitro differentiation potential. MSCs have multiple advantages including expression of minimal immunogenicity because of their low expression of major histocompatibility antigens, which in turn permits allogeneic therapy without immunosuppression. They can be expanded in vitro while maintaining an undifferentiated state, which allows generating sufficient amounts for clinical use, as well as cryopreservation before clinical use. They potentially have multiple effects on the host immune response to injury, while maintaining or augmenting the host response to pathogens and facilitating tissue repair.

They can be isolated from a variety of tissues, including bone marrow, fat, cord blood and tissue, and placenta [55,56]. MSCs represent an attractive therapeutic tool in regenerative medicine due to their ability to enhance regeneration and repair, modulate immune response, promote angiogenesis, and protect tissues from injury [57,58].

The beneficial effects of MSCs or MSC-conditioned medium in rodent models of hyperoxia-induced BPD were shown by multiple studies using intravenous [59,60,61,62,63,64], intratracheal [65,66,67,68,69], intraperitoneal [70,71], and intranasal [72] routes of administration.

Treatment with MSCs in these animal models reversed BPD and associated pulmonary hypertension, prevented arrested alveolar growth, and restored lung alveolarization and vascularization [64,66,72,73]. MSC treatment also helped to improve pulmonary architecture, attenuate pulmonary fibrosis, and increase the survival rate of BPD mice [70], as well as lower the expression of the profibrotic factors angiotensin II, angiotensin II type 1 receptor, and angiotensin-converting enzyme [65]. In addition, MSCs induced immune modulation, decreased inflammation, and lowered levels of inflammatory mediators such as IL−6 and TNF-α [59,60]. The beneficial effects of MSCs are thought to be attributed to a concerted effort targeting angiogenesis, immunomodulation, wound healing, and cell survival [72].

### 5.1. MSC Engraftment and Improved Pulmonary Architecture in BPD Models

Early animal studies used bone-marrow-derived MSCs to prevent arrested alveolar growth and lung injury in BPD models of hyperoxia-induced lung injury in neonatal rats or neonatal mice [60,66]. MSCs improved survival and exercise tolerance, reduced alveolar loss and lung inflammation, decreased fibrotic changes, decreased expression of α-smooth muscle actin, and prevented pulmonary hypertension [60,62,66,68,69]. MSCs were shown to protect against neonatal hyperoxic lung injury through stimulation of vascular endothelial growth factor (VEGF) [63], as well as limit the downregulation of thyroid transcription factor-1 (TTF-1), which plays a key role in lung morphogenesis [67]. They also increased the expression of natriuretic peptide B (NPPB), a neovasculatory factor that stimulates endothelial regeneration [72].

Engrafted MSCs express the type 2 alveolar epithelial cell-specific marker surfactant protein C (SP-C) [66,68,70]. However, several studies found that engraftment was disproportionately low for cell replacement to account for the therapeutic benefit, suggesting a paracrine-mediated mechanism [60,66,69].

### 5.2. Therapeutic Effects of MSC-Secreted Products in BPD Models

Numerous studies found that treatment with MSC-conditioned media (MSCs-CM) had a comparable or even a more pronounced effect than MSCs themselves, preventing both vessel remodeling and alveolar injury [60,61,64,67,69]. As little as a single dose of MSCs-CM treatment (1) reversed the hyperoxia-induced parenchymal fibrosis and peripheral pulmonary artery devascularization, (2) partially reversed alveolar injury, (3) normalized lung function (airway resistance, dynamic lung compliance), (4) fully reversed the moderate pulmonary hypertension and right-ventricular hypertrophy, and (5) attenuated peripheral pulmonary artery muscularization associated with hyperoxia-induced BPD [64]. MSCs and MSC-secreted products also enhanced recovery and repair following ventilator-induced lung injury by enhancing the restoration of systemic oxygenation and lung compliance, reducing total lung water, decreasing lung inflammation and histological lung injury, and restoring lung structure [59]. Daily administered intraperitoneal injection of MSC-derived exosomes protected alveolarization and angiogenesis in a hyperoxia-exposed neonatal rat model of BPD [71]. These studies further emphasize the paracrine role of MSCs as a treatment of BPD.

### 5.3. Anti-Inflammatory Role of MSCs in BPD Models

Studies have shown that MSCs or MSC-derived factors have both anti-inflammatory and proangiogenic mechanisms to protect the lung from hyperoxia-induced lung and heart disease associated with BPD [71]. Animals treated with MSCs-CM had normal alveolar numbers and drastically reduced lung neutrophil and macrophage accumulation. Macrophage stimulating factor 1 and osteopontin, both present at high levels in MSCs-CM, may be involved in this immunomodulation [60]. MSCs reduced the production of the proinflammatory cytokines TNFα, CX3CL1, IL-6, IL-1β, and TIM-1, while they increased the concentration of IL-10 [59,65,67,71,72].

### 5.4. MSC Effects on Alveolar Apoptosis and Angiotensin System in BPD Models

The renin–angiotensin system (RAS) is believed to play a role in neonatal lung development and BPD pathogenesis including septal formation, oxygen-induced inflammation, and signaling for pulmonary alveolarization [2,74,75,76,77]. The RAS was shown to be involved in the apoptosis of alveolar epithelial cells, epithelial barrier function, and surfactant protein production in hyperoxia [78,79,80]. In addition to the immunomodulatory effects of MSCs, a study showed that human MSCs attenuate hyperoxia-induced lung injury through inhibition of the RAS in newborn rats. Hyperoxia induced activation of the profibrotic arm of RAS in newborn rat lungs. The rats reared in hyperoxia exhibited significantly higher expression of angiotensin II, angiotensin II type 1 receptor, and angiotensin-converting enzyme than those reared in room air. Administering MSCs to hyperoxia-exposed rats decreased expression of angiotensin II, angiotensin II type 1 receptor, and angiotensin-converting enzyme to normoxic levels [65]. Treatment with MSCs caused the suppression of alveolar cell apoptosis and lung inflammation responses to oxygen with upregulation of the expression of BCL-2 gene and downregulation of the expression of BAX gene [63].

### 5.5. Sources of Mesenchymal Stem Cells (MSCs) for BPD Animal Studies

MSCs were originally discovered in the bone marrow, and criteria for identifying them focused on the presence and absence of cell-specific markers such as CD105, CD90, and CD73, without the expression of CD45, CD34, CD14, CD11b, CD79a, CD19, or HLA-DR. MSCs have several advantages. First, they express minimal immunogenicity because they have low expression of major histocompatibility antigens, which in turn permits allogeneic therapy without immunosuppression. Second, they can be expanded in vitro while maintaining an undifferentiated state, and it is feasible to generate sufficient quantities for clinical use, as well as cryopreservation before clinical use. Third, they exert multiple effects on the host immune response to injury, while maintaining or augmenting the host response to pathogens and facilitating tissue repair [57,58].

#### 5.5.1. Bone-Marrow-Derived Mesenchymal Stem/Stromal Cells

Many research studies used mesenchymal stem/stromal cells (MSCs) because of their easy accessibility from a variety of sources such as bone marrow, muscle, adipose tissue, umbilical cord blood, Wharton’s jelly in the umbilical cord, peripheral tissue, and placenta. Various studies using MSCs sourced from bone marrow in a hyperoxia-induced model of neonatal lung injury have reported an improvement in the alveolar structure and prevention of alveolar growth arrest [60,61,62,63,64,66,67,68,70,71].

Furthermore, one study demonstrated that bone-marrow-derived MSCs administration also improves survival and overall lung function, prevents vascular growth arrest, attenuates lung inflammation, inhibits lung fibrosis, and reduces or reverses pulmonary hypertension [81]. Some of the limitations of bone-marrow-derived MSCs are the limited availability of bone marrow, the very painful and invasive procedure to obtain it, and the extremely low number and life span of the cells derived from the bone-marrow aspirate.

#### 5.5.2. Human Umbilical Cord Mesenchymal Stem/Stromal Cells (hUC-MSCs)

Umbilical cord blood (UCB) is a more viable and easier source to access with less invasive extraction procedures compared with bone marrow. Umbilical cord blood is a rich source of different mononuclear cell populations containing high levels of primitive, multipotent stem cells, progenitor cells, and immunoregulatory T cells [82]. The umbilical cord has been utilized to isolate different types of stem cells including perivascular cells, Wharton’s jelly MSCs, and UCB-derived MSCs [69,72]. Studies utilizing UCB-derived MSCs have shown similar results compared with the bone-marrow-derived MSCs in hyperoxia-induced models of neonatal lung injury. Some of the results demonstrated in the studies were improved alveolar structure and restoration of alveolar growth, attenuation of lung fibrosis, reduced lung inflammation, prevention of impaired lung angiogenesis, and improved exercise capacity. Compared with bone-marrow-derived MSCs, intratracheal administration of UCB-derived MSCs is easier to administer considering the lower minimum required dosage. A study to optimize the dose of human UCB-derived MSCs in attenuating hyperoxia-induced lung injury in newborn rats demonstrated that a dose of as little as 5 × 10^4^ MSCs was effective, identifying optimal protective effects with a dose of 5 × 10^5^ MSCs [83]. It has been observed that MSC administration is more advantageous in the early stages of lung development than in later stages [69,83,84,85].

While most animal studies on stem-cell therapy for BPD models utilized bone-marrow-derived MSCs [60,61,62,63,64,66,67,68,70,71], local administration of MSCs derived from human umbilical cord (hUC-MSCs) exhibited systemic effects. This was shown in a study in which intratracheal administration of hUC-MSCs was found to attenuate both lung and brain injuries in rat pups exposed to hyperoxia [86]. Other studies have shown that hUC-MSCs exert both short- and long-term therapeutic benefits without adverse lung effects in BPD experimental models using either single or multiple doses of hUC-MSCs [69,87]. The use of hUC-MSCs in BPD models showed restored lung alveolarization, vascularization, and pulmonary vascular remodeling. Specific effects are summarized as (i) anti-inflammatory effects, evident by the decrease in proinflammatory cytokines such as TGF-β, IFN-γ, macrophage MIF, and TNF-α, (ii) antifibrotic effects evident by the decrease in collagen density, MMPs, and elastin expression, and by the increase in VEGF, MMP-2, vessel density, and angiogenesis, and (iii) lung function improvement and accelerated repair evident by the decrease in BPD injury-related protein markers such as CX3CL1, TNF-α, TIM-1, hepassocin, neprilysin, osteoprotegerin, and LIF, and by the increase in the alveolar septal width and septal crest density [44,72,88,89,90].

#### 5.5.3. Placental Mesenchymal Stem/Stromal Cells (P-MSCs)

Although not as widely utilized, a couple of studies showed that P-MSCs derived from the placental tissues exhibited anti-inflammatory and antifibrotic effects in rodent models of lung injury and inflammation [91,92]. Lung function improvement and restored vascular density were noted with P-MSC treatments. P-MSCs showed a decrease in the proinflammatory cytokines IL-6 and TNF-α, a decrease in fibrosis markers, CTGF, collagen density, infiltrating macrophages, and neutrophil infiltration, and an increase in the angiogenic factor VEGF [44,91,92].

## 6. Other Sources of Stem-Cell Therapy for Prevention of Lung Injury

Human amniotic epithelial cells (hAECs) are isolated from amnion membranes of full-term delivered placenta and display features of embryonic and multipotent stem cells. These human stem-like cells can differentiate into lung epithelium, in addition to reducing inflammation and abrogating fibrosis post lung injury [93]. Earlier animal models on bleomycin-induced lung injury and fibrosis in mice showed that hAECs could reduce acute inflammation, decrease collagen density and fibrosis, accelerate repair, and improve lung function [93,94,95]. Later, studies in mouse or sheep models of BPD showed that administration of hAECs produced anti-inflammatory effects by decreasing the infiltration of inflammatory cells and the proinflammatory cytokines TNF-α, TGF-β, IFN-γ, PDGF-α, PDGF-β, IL-1β, IL-10, and IL-6. In addition, hAECs exhibited antifibrotic effects and decreased collagen density and peripheral pulmonary arterial remodeling, resulting in lung function improvement. The hAEC treatment also restored the alveolar architecture, improved secondary septal crest density and lung tissue-to-air space ratio, and increased the density of pulmonary capillary bed, promoting pulmonary angiogenesis [96,97,98,99].

## 7. Human Clinical Trials of Stem Cells for Prevention of BPD

Undoubtedly, in recent years, there seems to have been unbounded interest concerning mesenchymal stem cells (MSCs). This is attributed to their exciting characteristics including long-term ex vivo proliferation, multilineage potential, and immunomodulatory properties. In this regard, MSCs emerge as attractive candidates for various therapeutic applications [81]. Several clinical trials are currently ongoing testing the safety and efficacy of using MSCs in BPD patients [100]. So far, three of the completed translational trials published results on the use of MSCs for treating BPD in humans [101,102,103]. Two phase I dose-escalation studies tested the safety and feasibility of intratracheal MSCs in preterm infants at high risk of BPD. The treatment was well tolerated and without any serious adverse effects. As a secondary endpoint, Chang et al. also reported some preliminary evidence of the treatment’s efficacy, reporting lower levels of IL-6, IL-8, MMP-9, TNF-α, and TGF-β1 in tracheal aspirates after MSC treatment, and MSC recipients seemed to have less severe BPD. A two2 ear follow-up study confirmed the lack of any long-term side-effects in babies treated with MSCs [104]. However, these trials identified the need for a larger, blinded randomized clinical trials.

A phase II double-blind, randomized, placebo-controlled clinical trial was conducted on preterm infants at 23 to 28 gestational weeks receiving mechanical ventilatory support with respiratory deterioration [101]. The study was conducted on 33 premature infants- receiving hUC-MSCs or premature infants receiving placebo. Although the inflammatory cytokines in the tracheal aspirate fluid were significantly reduced with MSC treatment, the primary outcome of death or severe/moderate BPD in the control group was not significantly improved with MSC transplantation. However, the severity BPD was significantly improved from 53% to 19% with MSC transplantation in the 23 to 24 gestational weeks group. The authors of the clinical trial study reported that the study was underpowered to detect its therapeutic efficacy in preterm infants [101]. One trial used extracellular vesicles (EVs) for preterm neonates at risk for BPD. The source used in this trial was UNEX-42, a preparation of EVs secreted from human bone-marrow-derived MSCs. However, this trial was terminated due to a business decision [105].

## 8. Paracrine Effect of MSC Therapy in BPD

Therapeutic actions of MSCs have been hypothesized to be related to their immunomodulatory/anti-inflammatory/angiogenic/antibacterial effects and to their regenerative effect in injured tissue [106]. In animal models of BPD, MSCs were shown to increase the number of bronchioalveolar stem cells and distal epithelial progenitor cells in the lung by paracrine signaling [61]. Few engrafted human MSCs have been detected in BPD animal models, according to immunofluorescence analysis and human-specific sequencing [69]. Therefore, the effects of MSCs in BPD animal models must be explained by the repair process that functions via paracrine effects, rather than by the engraftment and differentiation of exogenous MSCs [106].

### 8.1. Extracellular Vesicles (EVs)

After studying the role of MSCs in the regeneration of different tissues, studies have shown that the effect of MSCs in tissue regeneration is most likely due to paracrine action that stimulates different immune mechanisms and regenerative actions in vivo that enhance tissue repair. Recent studies showed that extracellular vesicles (EVs) play an essential role in this process.

EVs are a group of nanosized membrane-enclosed natural lipid bilayer vesicles secreted by every cell type in the body. EVs play a major role in intercellular communication by transferring cellular content between the cells, and they perform several functions in the cell microenvironment and extracellular spaces. Thus, they can be isolated from all body fluids and cell culture supernatants. These vesicles carry their bioactive cargo of different cellular components such as mRNA, miRNA, circular RNAs, lipids, and proteins. They also carry functional proteins on their surface. EVs through this cargo modulate cellular processes such as migration, proliferation, coagulation, inflammation, apoptosis, and angiogenesis. EVs can be classified according to their size and biogenesis. Currently, at least three main types of EVs are recognized: exosomes (30–150 nm in diameter), microvesicles (MVs) (also named microparticles or ectosomes; 0.1–1 μm in diameter), and apoptotic bodies (2–5 μm in diameter). Microvesicles are EVs released by direct budding through the cellular membrane. Exosomes are released by budding inside the endosomes to form multivesicular bodies (MVBs). These MVBs either fuse with the lysosomes for their content to be digested or fuse with the cell membrane for their content to be released outside the cell, including exosomes. Despite being biologically very distinct regarding their origin, there is no consensus in the scientific community about the distinction between them regarding their method of isolation and characterization. However, MVs tend to be larger than exosomes. In practice, some authors tend to name them smaller and larger EVs [107]. The most important advantages of exosomes over cell therapy are stability, efficiency after systemic delivery, immune tolerability, ability to overcome microvascular plugging, and feasibility to be loaded with therapeutic components to enhance efficacy.

### 8.2. Role of Extracellular Vesicles (EVs) in Mediating the Paracrine Effect of MSCs in Hyperoxia-Induced Lung Injury and BPD

The administration of EVs isolated from MSCs of different sources has been effective in the alleviation of the effects of hyperoxia in experimental newborn animals. Willis et al. compared the effect of exosomes isolated from Wharton-jelly-derived MSCs (WJMSCs) with those isolated from bone-marrow-derived MSCs (BMSCs) and human dermal fibroblasts [108]. The results showed that exosomes from both types of MSCs alleviated the long-term destructive effect of hyperoxia on lung tissues in the mouse models. They improved respiratory functions, ameliorated pulmonary hypertension, and reduced fibrous and vascular remodeling of lung tissue. This occurred through suppressing the proinflammatory alveolar macrophages (M1) and enhancing the anti-inflammatory subset (M2). A more comprehensive study by the same group showed that these exosomes interacted with the lung-associated myeloid cells to suppress the inflammatory reactions in response to hyperoxia. This led to alleviation of the hyperoxia-induced lung injury [109]. A similar study showed that WJMSCs alleviated the effect of hyperoxia in newborn rat models of BPD. However, they suggested a different mechanism through in vitro study showing that the WJMSCs inhibited the trans-differentiation of AT2 to AT1 [110]. Another study suggested the role of adipose-tissue-driven MSC EVs through combating oxidative stress induced by hyperoxia. Those EVs carried miR-21-5p, which suppressed the expression of SKP2 protein, leading to the activation of Nrf2 that enhanced the expression of antioxidant proteins in the cell [111]. An additional mechanism of action of the BMSC-EVs is the transfer of VEGF, which is essential for the normal blood vessels growth and relieves vascular abnormalities of the BPD [71]. The same mechanism was suggested for hUC-MSC-derived EVs, which additionally boost cellular regeneration through augmenting pAKT levels and inhibiting PTEN [112]. A summary of the role of MSC-secreted EVs is illustrated in Figure 1.

One of the advantages of EVs is the easier intratracheal administration compared with MSCs. A few recent studies showed positive results when administrating EVs intratracheally in animal models of BPD [111,112,113,114,115]. Some of these studies compared the effect of the EVs with their source MSCs. In all of these studies, both the EVs and their cells of origin alleviated the effect of hyperoxia on neonatal lungs. However, some of these studies found that the effect of both MSCs and their EVs was equal [113,115]. Contrary to these results, in the study by Li et al. [116], the authors concluded that MSCs were more effective than their EVs. One of the main differences between the two studies was the multiple application of treatment in Porzionato et al.’s study on postnatal days 3, 7, and 10, while, in Li et al.’s study, the treatment was applied only on postnatal day 7. Further work by Porzionato et al. confirmed the effectiveness of the administration of MSCs-EVs in ameliorating the long-term effects of hyperoxia on the lungs of newborns [114]. Previous studies also showed the intratracheal administration of MSC exosomes to be an effective therapy for other lung conditions such as bronchial asthma [117], fibrosis [118], sepsis [119,120] ischemic/reperfusion injury [121], and injury from cigarette smoking [122].

Another important advantage of using EVs is engineering or modifying them to get extra benefits from the treatment. The modification could include loading them with an extra gene, mRNA, miRNA, or protein. A recent study showed that mmu_circ_0001359 loaded on exosomes would attenuate the allergic symptoms in a mouse asthma model [123]. Similarly, microvesicles carrying miR-223/142 attenuated the LPS-induced inflammation [120]. Furthermore, the surface of the EVs could be modified to direct them toward specific cells for a better response or fewer side-effects. For example, EVs with RPB protein embedded into its surface, loaded with curcumin, and administered intranasally in rats induced with LPS alleviated the LPS-induced inflammatory response [124].

An additional advantage of EVs is the possibility of freeze-drying them (lyophilization) and reconstituting without losing much of their biological activity. Recent efforts have been directed toward inventing an approach that would best keep the biological activity of EVs. Lyophilization would enable a more extended shelf life of the EVs and safe transfer [125].

## 9. Challenges and Limitations for the Use of MSC-Based Therapies for BPD in Clinical Settings

From the foregoing discussion of the therapeutic efficacy of MSCs for the treatment of BPD in newborns, this cell-based approach is highly promising for use in clinical practice. However, there are many questions that need to be addressed before they can become an acceptable approach in the clinic. The first question that needs to be carefully evaluated is which source of MSCs is ideal for ease of procurement with strict adherence to ethical considerations. As mentioned elsewhere in this review, MSCs can be obtained from various sources such as bone marrow, umbilical cord blood, placenta, Wharton’s jelly, and adipose tissue. There have been few studies that compared MSCs from these sources with respect to their therapeutic efficacy against BPD. In terms of ease of procurement of tissue, placenta and adipose tissue may be a better source. A second consideration not addressed is the establishment of standardized conditions such as culture media composition and oxygen concentration of the growth chambers in culturing MSCs from these sources, as well as the effect of various culture conditions on the therapeutic effect of MSCs. Thirdly, for cell-based therapies, it is important to have a steady supply of large quantities of cells that are continuously consistent in their efficacy. Since it is not possible to generate cells in large numbers to meet clinical requirements at one time, it will be challenging to maintain consistent performance when the cells are obtained from different sources and at different times. Thus, it is critically important to standardize culture technologies that produce cells with the same efficacy every time. Another challenge to MSC-based therapy is determining whether the therapeutic potency of MSCs remains consistent when cells are passaged for the expansion of cell number. While it has been well documented that MSCs proliferate actively and can be expanded when passaged, it is not defined with clarity whether their therapeutic efficacy is altered or maintained during serial passages. This needs to be addressed to standardize the limit to which MSCs can be passaged in culture to expand their number without any change in their therapeutic potency. A critical limitation of cell-based therapies is the concern about their tumorigenic potential. It has been shown that embryonic stem cells and induced-pluripotent stem cells (iPSCs) could form teratomas in animal models. This raises questions about the safety of MSCs in regenerative medicine. Are MSCs nontumorigenic when administered to BPD patients? This question has not yet been clearly answered, but some studies suggested that MSCs are far less tumorigenic than ESCs or iPSCs [126,127,128], while one study implicated the tumorigenicity of MSCs [129]. A recent study evaluated the role of hUC-MSCs in tumorigenesis in an NOD SCID mouse model. The authors reported that hUC-MSCs were not tumorigenic and did not significantly promote or inhibit solid or hematological tumor growth or metastasis in NOD SCID mice [130]. The weight of evidence and the lack of immunogenicity of MSCs, however, suggest that MSCs may not pose a tumorigenicity risk.

## 10. Conclusions

While stem-cell and stem-cell-derived EV therapy could be effective therapeutic approaches for the treatment of BPD, some of the practical limitations of using stem-cell-based therapy in clinical practice need to be addressed. For clinical use, stem cells and stem-cell-derived EVs should have standardized technologies for isolation and preservation. They should be characterized for their uniform response, which may vary depending on the tissue of their isolation. This is important in the future for them to be used as promising biotherapeutics. The current preservation techniques for the long-term storage of stem cells and EVs need to be evaluated and improved to maintain their efficacy for BPD therapy.

## Figures and Tables

**Figure 1 cells-11-01275-f001:**
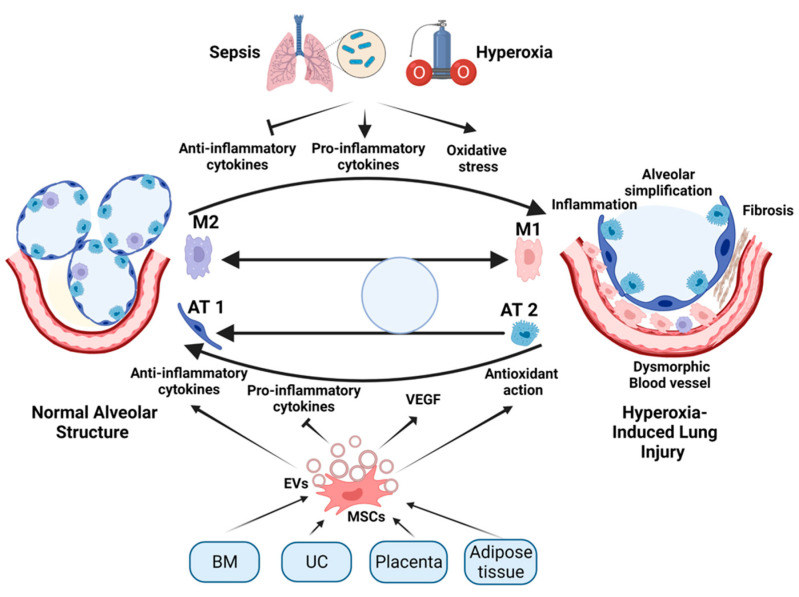
Schematic diagram illustration of the role of MSCs and secretion of their EVs in the treatment of hyperoxia-induced lung injury. As the preterm neonatal lung is exposed to excessive oxygen supplementation (hyperoxia) or infection (sepsis), proinflammatory cytokines are released, anti-inflammatory cytokines are inhibited, and oxidative stress occurs. These changes direct the conversion of M2 to M1 macrophage subsets, leading to structural lung damage/developmental abnormalities. MSCs are derived from various sources including bone marrow (BM), umbilical cord (UC), placenta, and adipose tissue. Treatment with MSCs or their secreted EVs alleviates the hyperoxia-induced lung injury through several mechanisms. These mechanisms include inhibition of proinflammatory cytokines and induction of anti-inflammatory cytokines, VEGF, and antioxidant pathways, leading to the transition of M1 to M2 macrophages, and the stimulation of differentiation of AT2 alveolar epithelial cells to AT1 alveolar epithelial cells. This illustration was created using BioRender.com.

## Data Availability

All data and material used for writing the manuscript are available.

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
