# Peer review of "Stem-Cell Therapy for Bronchopulmonary Dysplasia (BPD) in Newborns"

_cells, 2022, doi:10.3390/cells11081275_

Round 1
Reviewer 1 Report
I am very honored to review this fine manuscript from Michigan State University. This review focused on the use of mesenchymal stem cells (MSCs) of the treatment for bronchopulmonary dysplasia (BPD), and they not only stated how MSCs, MSCs-derived extracellular vesicles (EVs) are effective, but also a promising therapeutic modality for the treatment of BPD. All readers could agree that increasing more feasible treatment options in BPD is essential for improving the overall quality of life of preterm infants. Readers also may be convinced by this review that treatment with MSCs is a potential treatment for BPD. This manuscript is well written and will be useful for pediatricians who are responsible for many BPD patients. However, I have several concerns regarding this manuscript.
Major comments
- One of my concerns is the novelty of this review. There have been a large number of reviews as well as reports on newborn BPD using MSCs. Where is the novelty and originality of your review compared to those of MSC treatments published so far? The authors have referred to numerous reports on MSC treatment for infants and have analyzed and summarized those reports in detail. These detailed summaries are commendable. However, in order to publish this review manuscript, I think it is necessary to clearly state the differences from the previous reviews.
- The authors also briefly addressed the practical limitations of cell-based therapies in clinical practice. However, I think the reader's interest is in the detailed and clear answer to the limitations of these therapies in clinical practice. I understand that it is difficult for the authors to draw strong conclusions from the reports of a limited number of the clinical trials, but I think it is important to give a more clear opinion and suggestions for the authors on the limitations of the MSC treatment in clinical settings.
- Furthermore, I think it is necessary to state the potential disadvantages of this MSC treatment itself to patients, such as the fact that MSCs may stimulate cancer cells in vivo and in vitro. I agree that there is no doubt about the future potential of MSC treatment and its capabilities. However, it is important to properly summarize and clarify the potential disadvantages of this MSC treatment in current studies. I think this manuscript emphasizes the usefulness of MSC treatment but lacks some explanation of the potential disadvantages of this MSC treatment.
Minor comments
- The authors' comments should be removed from the PDF for review.
Author Response
Comments and Suggestions for Authors
I am very honored to review this fine manuscript from Michigan State University. This review focused on the use of mesenchymal stem cells (MSCs) of the treatment for bronchopulmonary dysplasia (BPD), and they not only stated how MSCs, MSCs-derived extracellular vesicles (EVs) are effective, but also a promising therapeutic modality for the treatment of BPD. All readers could agree that increasing more feasible treatment options in BPD is essential for improving the overall quality of life of preterm infants. Readers also may be convinced by this review that treatment with MSCs is a potential treatment for BPD. This manuscript is well written and will be useful for pediatricians who are responsible for many BPD patients. However, I have several concerns regarding this manuscript.
Major comments
- One of my concerns is the novelty of this review. There have been a large number of reviews as well as reports on newborn BPD using MSCs. Where is the novelty and originality of your review compared to those of MSC treatments published so far? The authors have referred to numerous reports on MSC treatment for infants and have analyzed and summarized those reports in detail. These detailed summaries are commendable. However, in order to publish this review manuscript, I think it is necessary to clearly state the differences from the previous reviews.
Authors response:
We appreciate the comment by the reviewer. We agree that there are numerous review papers in this subject, our review contains more up to date summary of the literature. Moreover, we expand in the role the MSCs paracrine effects, EVs and the placental derived MSCs and EVs because of high probability of more research in this area in prevention of lung injury and BPD.
,
- The authors also briefly addressed the practical limitations of cell-based therapies in clinical practice. However, I think the reader's interest is in the detailed and clear answer to the limitations of these therapies in clinical practice. I understand that it is difficult for the authors to draw strong conclusions from the reports of a limited number of the clinical trials, but I think it is important to give a more clear opinion and suggestions for the authors on the limitations of the MSC treatment in clinical settings.
Authors response:
The authors agree with the reviewer. More discussion was added regarding the limitations of cell- based therapy including BPD to the manuscript. See Section: 8
- Furthermore, I think it is necessary to state the potential disadvantages of this MSC treatment itself to patients, such as the fact that MSCs may stimulate cancer cells in vivo and in vitro.
Authors response:
The authors are in agreement with the reviewer, we added a section about the possible disadvantage of MSCs including tumorigenicity. Section:8
- I agree that there is no doubt about the future potential of MSC treatment and its capabilities. However, it is important to properly summarize and clarify the potential disadvantages of this MSC treatment in current studies. I think this manuscript emphasizes the usefulness of MSC treatment but lacks some explanation of the potential disadvantages of this MSC treatment.
Authors response:
The authors agree with the reviewer. The theoretical disadvantage of stem cells and MSCs and possible tumorigenicity make the use of EVs and exosomes a possible and more attractive alternative if future research studies support an equivalent effect in prevention of lung injury and development of BPD. This was added to the manuscript. Sections: 7 and 8
Minor comments
- The authors' comments should be removed from the PDF for review.
done
Reviewer 2 Report
General comments:
In recent decades, more and more premature infants have been born with a higher chance of survival due to improved neonatal care. However, BPD remains one of the major complications of preterm birth. So far, there is no curative therapy for BPD. Therefore, stem cell therapy is of great interest as a potential promising therapeutic approach for BPD.
In the present review the authors give an overview about the stem cell therapy in BPD by presenting human and animal data. The authors are summarizing the positive effect of different types of stem cells leading to morphological and functional improvement of the lung in BPD. They focus on mesenchymal stem cells (MSC) and stem cell-derived extracellular vesicles. They also discuss different sources of stem cells and briefly explain clinical studies and their limitations.
Thus, they conclude that stem cell therapy, if standardized, could be a potential treatment for BPD.
Major comments:
- The review present data from animal and human studies. It is sometimes difficult to follow whether the author is talking about human or animal data. (e.g. L102-107). Please correct.
- The description of the studies is kind of superficial. Please be more precise in terms of known or hypothesized mechanisms of action of MSCs. What are exactly EVs? What is inside? How does it work?
- Abstract, introduction and L102-107: ALI/ARDS in adult humans is not the same as BPD in neonates! Please be precise. I would recommend to rewrite this passage.
- I would recommend to restructure the review. Also, a general introduction to stems cells would be helpful.
Example 1: L242-251 (section 5.1) is not about clinical trials in BPD. The data referred to are generated using animal models.
Example 2: The title for section 5.1 is “Placental derived-MSCs therapy…” but the text does not give any information about that. Also, section 5.1 is associated with section 5 entitled “Clinical trials of stem cells in BPD” but did not give any information about clinical trials.
Example 3: Title of section 3 does not reflect the content of the subsections 3.1, 3.2, etc.
- In section 2 the authors mainly focus on the inflammatory mechanism as a risk factor for development of BPD but there is more than just inflammation, for example baro-/volutrauma by mechanical ventilation, dystrophy of the neonates...
- A section about the current limitations of stem cell therapies would be informative in this context .
Minor comments:
- Manuscript should be checked for typos, e.g. short and long-term (L37), Rodents (L70), The (L231). Also check for consistency. Sometimes the authors use “Stem/Stromal Cells”, sometimes “Stromal/Stem Cells”.
- The sentence L233/234 is not necessary. hUC-MSC was already mentioned before.
- All abbreviations should be explained (e.g. L69).
- Please check the illustration, e.g. the arrows should have the same distance to the writing
Author Response
Reviewer 2 Comments
Comments and Suggestions for Authors
General comments:
In recent decades, more and more premature infants have been born with a higher chance of survival due to improved neonatal care. However, BPD remains one of the major complications of preterm birth. So far, there is no curative therapy for BPD. Therefore, stem cell therapy is of great interest as a potential promising therapeutic approach for BPD.
In the present review the authors give an overview about the stem cell therapy in BPD by presenting human and animal data. The authors are summarizing the positive effect of different types of stem cells leading to morphological and functional improvement of the lung in BPD. They focus on mesenchymal stem cells (MSC) and stem cell-derived extracellular vesicles. They also discuss different sources of stem cells and briefly explain clinical studies and their limitations.
Thus, they conclude that stem cell therapy, if standardized, could be a potential treatment for BPD.
Major comments:
- The review present data from animal and human studies. It is sometimes difficult to follow whether the author is talking about human or animal data. (e.g. L102-107). Please correct.
Authors response:
The authors appreciate the comment by the reviewer. The manuscript was revised, and the data was clarified and made it more clear and separated the summary of the animal from human studies.
- The description of the studies is kind of superficial. Please be more precise in terms of known or hypothesized mechanisms of action of MSCs. What are exactly EVs? What is inside? How does it work?
Authors response:
In response to the reviewer’s comment, more details regarding the studies were added and more details regarding the mechanism of action of MSCs and EVs and more detailed overview of EVs.
- Abstract, introduction and L102-107: ALI/ARDS in adult humans is not the same as BPD in neonates! Please be precise. I would recommend to rewrite this passage.
Authors response:
As recommended by the reviewer, this section was corrected.
- I would recommend to restructure the review. Also, a general introduction to stems cells would be helpful.
Authors response:
As recommended by the reviewer, the manuscript was restructured, and more details were added regarding introduction to MSCs and EVs and more details regarding the animal and human trials.
Example 1: L242-251 (section 5.1) is not about clinical trials in BPD. The data referred to are generated using animal models.
Authors response:
As recommended by the reviewer, the section was revised and the above was corrected as suggested by the
Example 2: The title for section 5.1 is “Placental derived-MSCs therapy…” but the text does not give any information about that. Also, section 5.1 is associated with section 5 entitled “Clinical trials of stem cells in BPD” but did not give any information about clinical trials.
Authors response:
The manuscript was revised and the above was corrected as suggested by the reviewer.
Example 3: Title of section 3 does not reflect the content of the subsections 3.1, 3.2, etc.
- In section 2 the authors mainly focus on the inflammatory mechanism as a risk factor for development of BPD but there is more than just inflammation, for example baro-/volutrauma by mechanical ventilation, dystrophy of the neonates...
Authors response:
The manuscript was revised and the above was corrected as suggested by the reviewer.
- A section about the current limitations of stem cell therapies would be informative in this context .
Authors response:
This was corrected as suggested by the reviewer.
Minor comments:
- Manuscript should be checked for typos, e.g. short and long-term (L37), Rodents (L70), The (L231). Also check for consistency. Sometimes the authors use “Stem/Stromal Cells”, sometimes “Stromal/Stem Cells”.
Authors response:
This was corrected as suggested by the reviewer.
- The sentence L233/234 is not necessary. hUC-MSC was already mentioned before.
Authors response:
This was corrected in the revised manuscript as suggested by the reviewer.
- All abbreviations should be explained (e.g. L69).
Authors response:
This was corrected as suggested by the reviewer.
- Please check the illustration, e.g. the arrows should have the same distance to the writing
Authors response:
The illustration was revised and the distance of arrows was fixed in the revised manuscript as suggested by the reviewer.
Round 2
Reviewer 2 Report
Section 5.5.4. Human Amniotic Epithelial Cells (hAECs) does not fit to Section 5.5 since hAECs are not mesenchymal cells.
Thank you for having done the revision thoroughly.
Author Response
March 29, 2022
Re: Manuscript: Cells 1620432
Response to the reviewers’ comments
Reviewer 2 Comments
Below are the authors’ response to the reviewers/s comments in bold as reflected in the submitted revised manuscript.
Section 5.5.4. Human Amniotic Epithelial Cells (hAECs) does not fit to Section 5.5 since hAECs are not mesenchymal cells.
Thank you for having done the revision thoroughly.
Section 5.5.4. Human Amniotic Epithelial Cells (hAECs) does not fit to Section 5.5 since hAECs are not mesenchymal cells.
Thank you for having done the revision thoroughly.
Authors response:
We appreciate the comment by the reviewer. The authors agree with the reviewer suggested changes.
Section 5.5.4. Human Amniotic Epithelial Cells (hAECs) was removed from section 5.
The following section 6 was created as a new section with new title.
Please see the revised section with the new title.
6. Other sources of Stem Cells Therapy for Prevention of Lung Injury